# Knowledge, attitude, and practice towards Hepatitis B and vaccination status of pre-clinical medical students at Sylhet Women's Medical College, Bangladesh

Ramesh Lamichhane[1]*, Pritha Adhikari[2], Bishnu Deep Pathak[3], Aashika Rai[4], Pratikshya Ojha[4], Kripa Maharjan[4], Hamida Sultana Ruche[4], Madhusudan Saha[4]

1 Department of Gastroenterology and Hepatology, Jalalabad Ragib Rabeya Medical College, Sylhet, Bangladesh, 2 Department of Pediatrics, Nepalgunj Medical College, Nepalgunj, Nepal, 3 Department of Gastroenterology and Hepatology, Nepalese Army Institute of Health Sciences, Kathmandu, Nepal, 4 Department of Gastroenterology and Hepatology, Sylhet Women's Medical College, Sylhet, Bangladesh

* Dr.rameshlamichhane@gmail.com

**Data Availability Statement:** The complete data is available in a repository at figshare.com. The URL has been attached herewith. It is fully accessible.

## Abstract

### Background

Hepatitis B is a global health problem with high morbidity and mortality. The risk of transmission is more common among health care personnels and medical students during their professional health trainings. Vaccination is the most effective means of prevention. The main objective of this study was to determine the level of knowledge, attitude, and practice regarding Hepatitis B and vaccination among pre-clinical students in a medical college.

### Materials and methods

A web-based, single-center, descriptive cross-sectional study was conducted among pre-clinical medical students in Bangladesh from April 28, 2024, to May 4, 2024. The whole sampling technique method was used taking the entire population for our study. Data were collected using a self-administered questionnaire and analyzed using JMP Pro V17 Statistical Software.

### Results

Sixty-one (34%) students were vaccinated against Hepatitis B, of whom 18% received all three doses with a booster dose. The most common reason for non-vaccination was lack of awareness of one's vaccination status (43%). The median knowledge, attitude, and practice (KAP) scores were 54 (50–60), 19 (17–21) and 20 (19–23) respectively, and they were moderately positively correlated. Forty-six (25.98%) students had a good overall KAP score ($\geq$ 102).

### Conclusion

Only one-third of the students had been vaccinated, and the commonest reason for non-vaccination was lack of awareness of one's vaccination status. Nearly one-fourth of the

Figshare doi link: https://doi.org/10.6084/m9.figshare.25845664.v1.

**Funding:** The author(s) received no specific funding for this work.

**Competing interests:** The authors have declared that no competing interests exist.

participants had good knowledge, attitude, and practice related to Hepatitis B. Therefore, education regarding Hepatitis B infection, risk factors, and importance of vaccination is a must among pre-clinical medical students in Bangladesh.

## Introduction

Hepatitis B is a liver infection caused by a blood borne virus which is prevalent worldwide [1]. Hepatitis B virus (HBV) can cause both acute and chronic infections causing increased risk of liver cirrhosis and cancer related morbidity and mortality [2]. The global prevalence of chronic Hepatitis B is approximately 5%, with estimated 0.5–1.2 million deaths per year [1]. It is mainly transmitted through blood and other body fluids via different potential routes like blood transfusion, unsafe sexual contact, intravenous drug use and vertical transmission from mother to fetus [3–8].

The risk of acquiring Hepatitis B infection is found to be higher among health care personnels during their professional health trainings [4,9–11]. The incidence of such infection is found to be 2–4 times higher compared to the general population [4,12]. Similarly, medical students are more susceptible to HBV infection as they are exposed directly to infected patients, blood products, injections and surgical instruments during the course of their clinical rotations [2,4,12]. Additionally, lack of adequate experience during the early phase of clinical training puts health science students at higher risk of sustaining needle stick injuries; the prevalence of which varies between 14.6%-61.3% [3,4,13–15].

HBV vaccination, which provides around 90–100% protection, is the most effective way of preventing HBV infection, and is strongly recommended by World Health Organization (WHO) among high risk professions and academic programs [2,16]. For optimal prophylaxis, three doses of the vaccine at 0, 1 and 6–12 months are to be administered [5]. In addition, health care workers including medical students should have a proper knowledge of hepatitis B, so that they could adopt safety precautions, and educate others about the risks and preventive measures [4,5,12].

Hepatitis B is widespread in the Asia-Pacific region with a relatively higher prevalence (8–20%) in South East Asia [1]. Bangladesh is located in an intermediate endemicity zone where the prevalence of chronic HBV infection is 2–6% [17,18]. However, the vaccination rate among healthcare workers is comparatively lower in developing countries, including Bangladesh (18–39%) [2,16]. Medical students are the cornerstones of the health care system, and they need to be aware of the risks involved in dealing with their patients infected with Hepatitis B. Therefore, the primary objective of this study was to determine the level of knowledge, attitude, and practice regarding Hepatitis B, and vaccination status among pre-clinical medical students in Bangladesh.

## Material and methods

### Study design and setting

This descriptive cross-sectional study targeted pre-clinicalstudents (1st and 2nd year) at the Sylhet Women's Medical College in Bangladesh. The data were collected between April 28, 2024, and May 4, 2024, using a whole sampling method. The entire available population was used for sampling, resulting in 189 participants, including 89 first year and 100 second-year medical students. Participants were recruited through Google forms sent via a social media group

(Facebook, WhatsApp) by the class representative. At the start of the online survey, all participants were given the opportunity to express informed consent. This page provided thorough information regarding the study's background and objectives along with the investigators. The participants were informed that their participation was entirely voluntary, and they can wish to withdraw or participate in the study. Privacy and anonymity was ensured to prevent potential bias. No compensation was provided to the participants.

## Participants

**Inclusion criteria.**  All pre-clinical students from Sylhet Women's Medical College were recruited.

**Exclusion criteria.**  Students who were no longer attending class or could not be reached were excluded from the study.

## Study tools

We adapted a self-administered web-based questionnaire from the study by Shrestha DB et. al, 2020 with permission [3]. The original questionnaire assessed knowledge, attitude, and practice (KAP) related to Hepatitis B using a five-point Likert scale. It contained 25 items, including 10 for demographics, 5 for knowledge, 5 for attitude, and 5 for practice.

We made minor modifications to the questionnaire to suit our research objectives better. These modifications included assessing the history of Hepatitis B among participants' personal networks and their preferred mode of educational delivery. The original questionnaire demonstrated good internal consistency, with a Cronbach's alpha of 0.698. We assessed the internal consistency of the modified questionnaire using Cronbach's alpha. The alpha coefficient was 0.71, indicating acceptable internal consistency.

A five-point Likert scale was used to assess responses to questions about Hepatitis B knowledge, attitude, and practice, with 1 representing the least acceptable response and 5 representing the most acceptable response based on general understanding. Each question received a rating ranging from 1 (strongly disagree) to 5 (strongly agree). However, reverse scoring was used for some questions in the knowledge portion (questions 2d, 2f, 2g, 5b, and 5d) and the attitude section (questions 6 and 10). Knowledge of Hepatitis B was assessed using five major questions and additional sub-questions, yielding a potential score range of 15 to 75. The attitude and practice sections each had five questions, providing for a possible score range of 5–25. Overall total score was finally calculated by adding scores in individual section (i.e. knowledge, attitude and practice).

## Statistical methods

The data gathered from Google Forms were exported to Microsoft Excel-2021 and subsequently analyzed using JMP Pro V17 Statistical Software. Kolmogorov-Smirnov (K-S) and Shapiro-Wilk tests were used to evaluate the normality of the data distribution. The data were deemed normal if the significance value of the test was greater than 0.05, and non-normal if it was less than 0.05. Because our data were non-normal, non-parametric tests were used, with the median serving as a measure of distribution. A chi-square test or Fisher's exact test was performed to examine the relationship between variables, while Spearman's rho was used to assess the correlation between the total scores for knowledge and attitude, knowledge and practice, and attitude and practice. The quantitative data were presented as numbers along with the corresponding percentages.

### Ethical consideration

This study was approved by the Institutional Review Committee of Sylhet Women's Medical College (SWMC/Eth.C/IERB/202406).A consent form along with the objectives of our study was integrated into the questionnaire itself, and the students who consented to participate were directed to the questionnaire section. Those who did not consent were allowed to exit from the web-based form.

## Results

Of the 189 students targeted for the study, 177 were enrolled and analyzed, resulting in response rate of 93.6%. Among these, 82 (46%) were from the first year of MBBS and 95 (54%) were from the second academic year. The median age of the participants was 21 (20–21) years. The majority of students (58%) were aged 21 years and above. 22 (12%) were aware of Hepatitis B infections occurring within their personal networks, such as among friends, family, relatives, or close associations. The baseline characteristics and vaccination status of all the participants are presented in **Table 1**.

### Hepatitis B vaccination status and associated factors

Of all respondents, 61 (34%) reported that they had received the Hepatitis B vaccine. Of these, 31 (18%) students received three complete doses along with the booster dose, whereas 19 (11%) received three doses of the vaccine only. A mere 3% of them had received one or two vaccine doses, and the remaining 116 (66%) students had not received any dose of HBV vaccine. **(Fig 1A)** When asked why they had not been vaccinated, the most common responses were "lack of awareness of Hepatitis B vaccination status" (43%) and "lack of accessibility to the HBV vaccine" (28%). Additionally, one-fifth (21%) reported that they had no knowledge of the importance of the vaccine. **(Fig 1B)** In terms of preferred methods of education regarding Hepatitis B, interactive workshops, seminars (18%), and online resources such as videos and websites (15%) were the most popular among the respondents **(Table 1) (Fig 1C)**.

### Assessment of knowledge related to Hepatitis B

The median knowledge score was 54 (50–60) **(Table 3)**. Majority of the students agreed that Hepatitis B is caused by virus (53.1% agreed and 25.4% strongly agreed). Approximately one-third of the participants strongly agreed that Hepatitis B is transmitted from the mother to the fetus (33.9%), contaminated blood and body fluids (32.2%), unprotected sex with infected individuals (35.5%), and unsterilized syringes/needles (30.5%). Moreover, 42.9%, 19.2%, and 16.3% of respondents strongly disagreed with casual contact (shaking hands), coughing/sneezing, and contaminated food/water, respectively, as potential routes of transmission. Similarly, the majority of participants (49.7% agreed and 16.3% strongly agreed) knew that Hepatitis B can cause liver cancer, while one-fifth of the participants (22%) were neutral to this information. Likewise, 37.2% participants agreed and 21.4% strongly agreed on the fact that health care workers are at increased risk of getting Hepatitis B compared to general population. Regarding the prevention of Hepatitis B, most participants believed that vaccination is an effective means (50.2% agreed and 28.2% strongly agreed), unlike a few who denied the HBV vaccine as a preventive method (5.6% disagreed and 3.9% strongly disagreed). In the same way, 39.5% agreed and 21.4% strongly agreed that avoiding sharp needle and syringe injuries prevent Hepatitis B. To the contrary, some participants stated that the use of antivirals (35% agreed and 10.7% strongly agreed) and avoiding contaminated food/water (42.3% agreed and 15.2% strongly agreed) could prevent Hepatitis B. Likewise, most of the respondents agreed on

**Table 1. Baseline characteristics and vaccination status of the medical students.**

| Variables | | N | % |
|---|---|---|---|
| Age, years | 20 and below | 75 | 42 |
| | 21 and above | 102 | 58 |
| | Mean ± SD | 20.6 ± 1.0 | |
| | Median (IQR) | 21 (20–21) | |
| Academic year (MBBS) | 1st year | 82 | 46 |
| | 2nd year | 95 | 54 |
| Vaccination against Hepatitis B | Yes | 61 | 34 |
| | No | 116 | 66 |
| Dose of HBV vaccine received | Not vaccinated | 116 | 66 |
| | First dose | 5 | 3 |
| | Second dose | 6 | 3 |
| | Third dose | 19 | 11 |
| | Third dose along with the booster dose | 31 | 18 |
| Reason for not being vaccinated against Hepatitis B | Not aware of the Hepatitis B vaccine status. | 46 | 43 |
| | Lack of knowledge about the importance of Hepatitis B vaccine | 22 | 21 |
| | Lack of accessibility for Hepatitis vaccine. | 30 | 28 |
| | High cost of Hepatitis B vaccine | 6 | 6 |
| | Others | 3 | 3 |
| History of Hepatitis B infection among personal networks | Yes | 22 | 12 |
| | No | 155 | 88 |
| Preferred mode of education regarding Hepatitis B | Interactive workshops or seminars | 116 | 18 |
| | Online resources (e.g. videos, websites) | 99 | 15 |
| | Traditional lectures | 55 | 9 |
| | Printed materials (e.g. pamphlets, brochures) | 58 | 9 |

(HBV: Hepatitis B Virus; MBBS: Bachelor of Medicine and Bachelor of Surgery; SD: Standard Deviation).

using gloves when handling body fluids for the prevention (46.3% agreed and 32.2% strongly agreed) (**Table 2**).

## Assessment of attitude related to Hepatitis B

The median attitude score was 19 (17–21) (**Table 3**). Thirty-four (19.2%) participants strongly disagreed on feeling uncomfortable sitting with Hepatitis B infected person whereas 48 (27.1%) were neutral in their response. However, a few students (n = 6, 3.3%) indicated that they feel uncomfortable in these situations. About one-third of the participants did not mind shaking hands/hugging with an HBV-infected person (33.3% agreed and 10.1% strongly agreed). The majority of students believed that healthcare workers should receive Hepatitis B vaccination (33.9% agreed and 49.1% strongly agreed). Similarly, 78 (44%) agreed and 59 (33.3%) strongly agreed with the safety and effectiveness of the HBV vaccine. Likewise, most study participants acknowledged the need for HBV vaccination because they think they are at risk (51.9% strongly agreed) (**Table 2**).

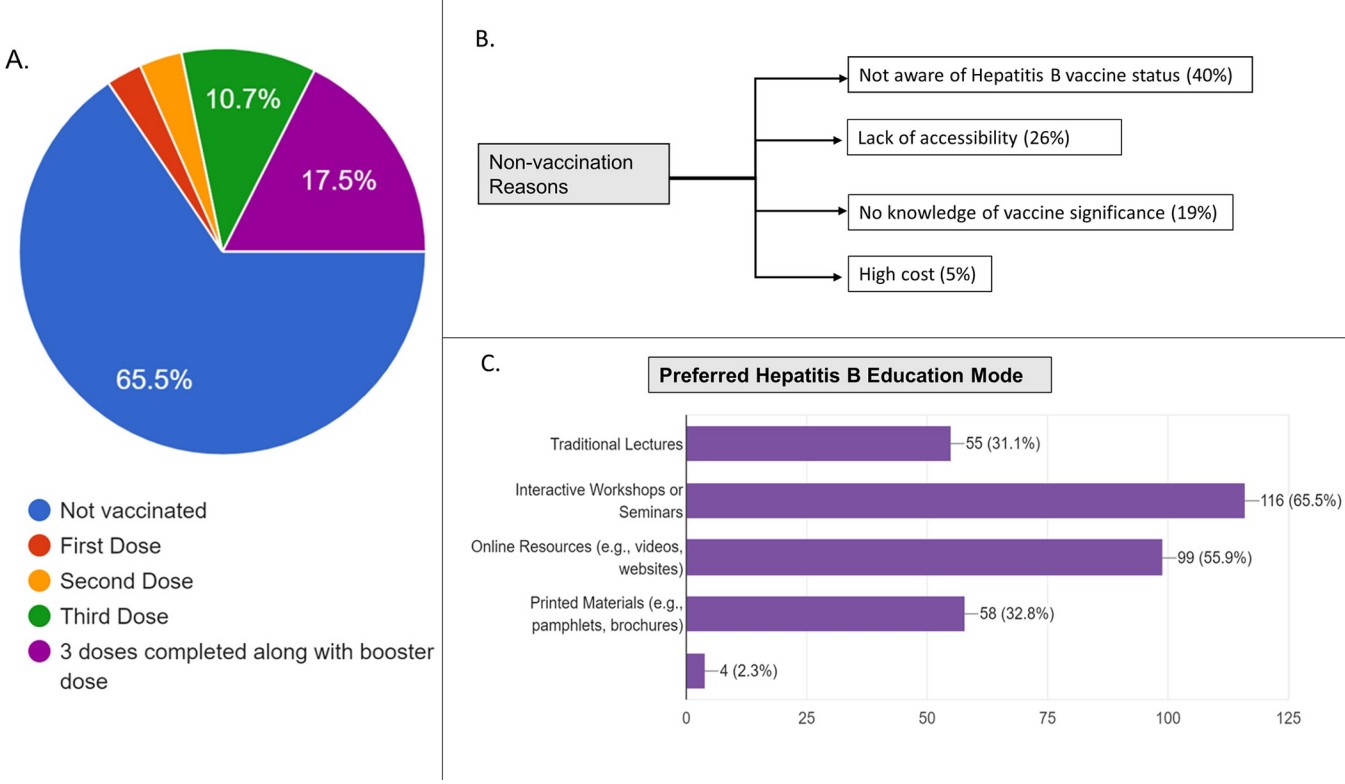

**Fig 1.** Illustrates distribution of vaccination doses received by participants (Panel A), factors contributing to non-vaccination (Panel B), and preferred modes of education on Hepatitis B within the study cohort (Panel C).

## Assessment of practice related to Hepatitis B

The median score of practice related to Hepatitis B was 20 (19–23) (**Table 3**). Most students (49.1% agreed and 32.2% strongly agreed) used a new blade for shaving or hair cutting. Likewise, 82 (46.3%) ask for a new syringe before injection, and 77 (43.5%) ask for sterilized equipment for ear/nose piercings. However, a small number of participants strongly disagreed with asking for a new syringe (n = 3, 1.6%) or sterilized piercing equipment (n = 4, 2.2%). Approximately one-fourth of the students strongly agreed on regular reporting of needle pricks or sharp injuries (n = 42, 23.7%), whereas four (2.25%) participants do not report such injuries. Furthermore, 44 (24.8%) respondents strongly agreed to attend the Hepatitis B related awareness program (**Table 2**).

## Correlation between knowledge, attitude, and practice scores

There was a moderate positive correlation between knowledge and attitude (r = 0.5319), knowledge and practice (r = 0.6173), and attitude and practice (r = 0.5473), all of which were statistically significant (p < 0.0001) (**Table 4**).

## Categorization of KAP score and its association with baseline characteristics

Out of total participants, 46 (25.98%) had a good KAP score (≥ 102). There was statistically significant association of KAP score category with vaccination status (p = 0.03), dose of HBV vaccine received (p = 0.009) and previously received information (p = 0.01) (**Table 5**).

**Table 2. Total knowledge, attitude, and practice scores distribution.**

| Question for response | Strongly disagree n (%) | Disagree n (%) | Neutral n (%) | Agree n (%) | Strongly agree n (%) |
|---|---|---|---|---|---|
| **Assessment of Knowledge related to Hepatitis B** | | | | | |
| 1. Hepatitis B is caused by virus. | 7(3.9) | 9(5.0) | 22(12.4) | 94 (53.1) | 45(25.4) |
| 2 Hepatitis B virus can be transmitted by | | | | | |
| a. Infected mother to fetus | 6(3.3) | 9(5.0) | 21(11.8) | 81 (45.7) | 60(33.9) |
| b. Contaminated blood & body fluids | 10(5.6) | 11(6.2) | 26(14.6) | 73 (41.2) | 57(32.2) |
| c. Unprotected sex with infected ones | 9(5.0) | 21(11.8) | 25(14.1) | 59 (33.3) | 63(35.5) |
| d. Casual contact (shaking hands) | 76(42.9) | 64(36.1) | 21(11.8) | 15(8.4) | 1(0.5) |
| e. Unsterilized syringes/needles | 9(5.0) | 12(6.7) | 24(13.5) | 78 (44.0) | 54(30.5) |
| f. Coughing/sneezing | 34(19.2) | 55(31.0) | 42(23.7) | 36 (20.3) | 10(5.6) |
| g. Contaminated food/water | 29(16.3) | 37(20.9) | 38(21.4) | 57 (32.2) | 16(9.0) |
| 3. Hepatitis B virus can cause liver cancer. | 6(3.3) | 15(8.4) | 39(22.0) | 88 (49.7) | 29(16.3) |
| 4. Health care workers are at increased risk of getting Hepatitis B than general population. | 16(9.0) | 16(9.04) | 41(23.1) | 66 (37.2) | 38(21.4) |
| 5. Hepatitis B can be prevented by | | | | | |
| a. Vaccination | 7(3.9) | 10(5.6) | 21(11.8) | 89 (50.2) | 50(28.2) |
| b. Antivirals | 13(7.3) | 20(11.3) | 63(35.5) | 62 (35.0) | 19(10.7) |
| c. Avoiding sharp needle/syringe injury | 11(6.2) | 12(6.7) | 46(25.9) | 70 (39.5) | 38(21.4) |
| d. Avoiding contaminated water/food | 12(6.7) | 19(10.7) | 44(24.8) | 75 (42.3) | 27(15.2) |
| e. Using gloves when handling body fluid | 8(4.5) | 5(2.8) | 25(14.1) | 82 (46.3) | 57(32.2) |
| **Assessment of attitude related to Hepatitis B** | | | | | |
| 6. I feel uncomfortable sitting with Hepatitis B infected person. | 34(19.2) | 55(31.0) | 48(27.1) | 34 (19.2) | 6(3.3) |
| 7. I don't mind shaking hands/ hugging with Hepatitis B infected person. | 25(14.1) | 28(15.8) | 47(26.5) | 59 (33.3) | 18(10.1) |
| 8. I believe Hepatitis B vaccine is safe and effective. | 5(2.8) | 7(3.9) | 28(15.8) | 78 (44.0) | 59(33.3) |
| 9. I believe health care workers should receive Hepatitis B vaccination. | 5(2.8) | 8(4.5) | 17(9.6) | 60 (33.9) | 87(49.1) |
| 10. I don't need Hepatitis B vaccination because I am not at risk. | 92(51.9) | 51(28.8) | 28(15.8) | 5(2.8) | 1(0.5) |
| **Assessment of practice related to Hepatitis B** | | | | | |
| 11. I use a new blade for shaving/hair cutting. | 5(2.8) | 4(2.2) | 24(13.5) | 87 (49.1) | 57(32.2) |
| 12. I ask for a new syringe before injection. | 3(1.6) | 7(3.9) | 12(6.7) | 73 (41.2) | 82(46.3) |
| 13. I ask for sterilized equipment for ear/nose piercings. | 4(2.2) | 4(2.2) | 22(12.4) | 70 (39.5) | 77(43.5) |
| 14. I always report for needle pricks / sharp injuries. | 4(2.2) | 6(3.3) | 33(18.6) | 92 (51.9) | 42(23.7) |

(*Continued*)

**Table 2.** (Continued)

| Question for response | Strongly disagree n (%) | Disagree n (%) | Neutral n (%) | Agree n (%) | Strongly agree n (%) |
|---|---|---|---|---|---|
| 15. I attend Hepatitis B related awareness program | 7(3.9) | 15(8.4) | 46(25.9) | 65 (36.7) | 44(24.8) |

**Table 3. Participants' responses related to knowledge, attitude and practice.**

|  | Knowledge sum(n = 177) | Attitude sum (n = 177) | Practice Sum (n = 177) | Total Score (n = 177) |
|---|---|---|---|---|
| Mean ± SD | 54.01±7.57 | 19.05±2.95 | 20.1±3.90 | 93.2±12.45 |
| Median (IQR) | 54 (50–60) | 19 (17–21) | 20 (19–23) | 93 (85.5–102) |

(IQR: Interquartile range; SD: Standard Deviation).

**Table 4. Correlation between knowledge, attitude, and practice scores related to Hepatitis B.**

| Variables | Spearman's correlation coefficient | p-value |
|---|---|---|
| Knowledge-Attitude | 0.5319 | < 0.0001 |
| Knowledge-Practice | 0.6173 | < 0.0001 |
| Attitude-Practice | 0.5473 | < 0.0001 |

## Discussion

Hepatitis B is a common and serious health problem which affects about more than 2 billion people worldwide [1,2]. Health care workers and medical students are the most vulnerable groups that warrant screening and vaccination against Hepatitis B [3,13]. Moreover, pre-clinical medical students are less experienced, and once they enter into their clinical phase, they are more likely to sustain needle-stick injuries while performing different procedures with infected patients. They remain in direct contact with the infected patients, their blood, blood products, and other surgical instruments which put them at higher risk of contracting Hepatitis B. Therefore, vaccination and knowledge regarding safety precautions are essential during the course of medical training [4,5,12,13].

In our study, 35% of the students were vaccinated against Hepatitis B. In Bangladesh, mass vaccination against hepatitis B was introduced into the Expanded Program on Immunization (2003–2005) schedule for infants, with three doses administered within the first year of life [19,20]. Yet, the rate of vaccination in our study was low, arguably due to phase wise implementation and potential unawareness about the vaccination status. The vaccination rate in this study is significantly lower compared to similar studies done in Nepal, where the vaccination rate was 86.5% [21] and 60.8% [3]. Likewise, a higher percentage of medical students received HBV vaccine in medical university/colleges of Saudi Arabia (69.5%) [22], Uganda (66.84%) [23], Pakistan (60.2%) [24], and Nigeria (47.7%) [25]. On the other hand, a similar study in India showed lower vaccination rates in a medical institution (26.7%) [26]. Moreover, the number of vaccinated students completing all three doses of vaccine was relatively higher in our case (81.97%) in comparison to those in Uganda (66.34%) [23], Nepal (60.9%) [3], Saudi Arabia (38%) [22], Pakistan (33%) [24], and Ethiopia (2%) [27]. Contrary to this, a study from Nepal by Bhattarai S et al reported similar findings, where the rate of full dose vaccination was 83.7% [21]. All these discrepancies could be due to different sampling size, sampling method and baseline population characteristics.

**Table 5. Association of the KAP Score category with baseline characteristics.**

| Variables | | KAP score category | | P-value |
|---|---|---|---|---|
| | | Inadequate (<102) n (%) | Good (≥102) n (%) | |
| Age | 20 or below (n = 75) | 53 (70.67) | 22 (29.33) | 0.39 |
| | 21 or above (n = 102) | 78 (76.47) | 24 (66.47) | |
| Academic year | 1st year MBBS (n = 82) | 60 (73.17) | 22 (26.83) | 0.86 |
| | 2nd Year MBBS (n = 95) | 71 (74.74) | 24 (64.74) | |
| Vaccinated against Hepatitis B | Yes (n = 61) | 39 (63.93) | 22 (53.93) | **0.03** |
| | No (n = 116) | 92 (79.31) | 24 (20.69) | |
| Dose of Hepatitis B vaccine | Not vaccinated (n = 116) | 92 (79.31) | 24 (20.69) | **0.009** |
| | First Dose (n = 5) | 2 (40.0) | 3 (60.0) | |
| | Second Dose (n = 6) | 2 (33.33) | 4 (66.67) | |
| | Third Dose (n = 19) | 16 (84.21) | 3 (15.79) | |
| | Third dose complete (n = 31) with booster dose | 19 (61.29) | 12 (38.71) | |
| Reason for not being vaccinated against Hepatitis B | Lack of knowledge about the importance of Hepatitis B vaccine (n = 22) | 17 (77.27) | 5 (22.72) | 0.97 |
| | Lack of accessibility for Hepatitis B vaccine (n = 30) | 23 (76.67) | 7 (23.33) | |
| | Not aware of the Hepatitis B vaccine status (n = 46) | 34 (73.91) | 12 (26.09) | |
| | High cost of Hepatitis B vaccine (n = 6) | 5 (83.33) | 1 (16.67) | |
| | Others (n = 3) | 3 (100.0) | 0 | |
| Previously received information | Yes (n = 156) | 111 (71.15) | 45 (28.85) | **0.01** |
| | No (n = 21) | 20 (95.24) | 1 (4.76) | |
| Source of Information | Television (n = 13) | 11 (84.62) | 2 (15.38) | 0.67 |
| | Social media for example YouTube, Facebook (n = 44) | 31 (70.45) | 13 (29.55) | |
| | Awareness campaign program conducted in high school or any other events (n = 58) | 41 (70.69) | 17 (29.31) | |
| | Friends or family (n = 45) | 36 (86.67) | 9 (13.33) | |
| | Healthcare provider (n = 17) | 12 (70.59) | 5 (29.41) | |
| History of Hepatitis B infection among personal networks | Yes (n = 22) | 16 (72.72) | 6 (27.27) | 1.00 |
| | No (n = 155) | 115 (74.19) | 40 (35.81) | |

**Note:** Bold p-values are statistically significant (<0.05).

The most common reasons for not getting vaccinated in this study were lack of awareness about HBV vaccination status (43%), inaccessibility to the vaccine (28%), and lack of knowledge about the importance of vaccination (21%). However, studies from Nepal by Shrestha DB et al. (74.6%) [3] and Bhattarai S et al. (43.2%) [21] depicted the most common reason being lack of vaccination programs offered in their regions. Likewise, 33.7% medical students in Pakistan were not vaccinated because they believed that they were not at risk [28]. Moreover, a study from Uganda reported high cost of vaccination as the primary reason for non-vaccination (63.2%) [23]. In addition to our study, lack of awareness was also found to be a common cause of non-vaccination in Nepal (5.6%) [3], and Pakistan (23%) [24]. Therefore, awareness regarding the importance and accessibility of vaccination should be disseminated among medical students.

Our study consistently documented a lower level of knowledge regarding different aspects of Hepatitis B. With regard to the routes of transmission for this disease, the most common

routes reported by our participants were vertical transmission (79.66%), unsterilized needles/syringes (74.58%), contaminated blood and body fluids (73.45%), and unprotected sex (68.93%). However, relatively higher percentage of medical students from other countries knew about these transmission routes [3,22,26,27]. In India, 94.4% and 83.9% of students knew that Hepatitis B is transmitted through contaminated blood and blood products and unsafe sex, respectively [26]. Likewise, in a study by Shrestha DB from Nepal, most of the students had knowledge regarding potential routes of transmission of Hepatitis B, namely contaminated blood and body fluids (97.3%), mother to child (92.2%), unsterilized syringe/needles (95%), and unprotected sex (87.8%) [3]. Moreover, more than 90% participants from Saudi Arabia [22] and Ethiopia [27] agreed that unsterilized syringe, needles and medical equipments are potential transmission routes. In the same way, 66.1% participants in our study knew that Hepatitis B can cause liver cancer. However, the percentage of students aware of this fact was higher in studies from Ethiopia (81.4%) [27], Nepal (80.6%) [3], and Saudi Arabia (75.5%) [22].

In our study, approximately half of the pre-clinical medical students reported feeling comfortable and easy in sitting with HBV infected people (50.28%), and do not mind shaking hands and hugging them (43.50%). This was comparable to findings from Nepal, where 43.7% and 56.3% of students found it comfortable in sitting, shaking hands, and hugging infected patients respectively [3]. Similarly, 77.40% of our participants believed that the Hepatitis B vaccine is safe and effective. This finding was higher than the students of Saudi Arabia (63%) [29] but lower than those from Nepal (86.7%) [3] and Ethiopia (81.7%) [27]. Notably, a small proportion of study participants (3.39%) told that they do not need vaccination because they are not at risk of Hepatitis B. This is parallel to the findings from the studies of Nepal (3.9%) [3] and India (3.7%) [26].

Regarding practice related to Hepatitis B prevention, more than 80% of our participants ask for new blade for shaving/hair cutting, new syringe before injection, and sterilized equipment for ear/nose piercing. This was comparable to the findings from India [26] and Nepal [3], although the percentage of Nepalese students asking for new syringe before injection was reported to be slightly higher (96.2%) than ours (87.57%). Approximately three-fourth of our pre-clinical students always report for needle pricks orsharp injuries, which is higher than those from Nepal (64.6%) [3], and Ethiopia (53.7%) [27].

The present study demonstrated moderate positive correlation between knowledge, attitude, and practice related to Hepatitis B among pre-clinical medical students. This was in line with the findings of a study by Shrestha DB from Nepal [3] which showed a weak positive correlation. Likewise, a study from India showed weak correlation between knowledge and practice that was statistically significant [26]. Similarly, in our study, one-fourth of the participants had a good KAP score ($\geq$ 102), which was significantly associated with vaccination status, dose of vaccine received, and previously received information. However, no significant association was found with baseline variables in a study by Shrestha DB et al [3].

In a nutshell, our study demonstrated a very low vaccination rate in pre-clinical medical students of Bangladesh with most common cause being lack of awareness of vaccination status. This finding has several clinical implications. Firstly, it indicates that the pre-clinical medical students are at increased risk of Hepatitis B infection and there is an immediate need for enhanced vaccination programs. Lack of awareness of one's vaccination status depicts a significant gap in the current vaccination programs and protocols within the medical institutions as well as insufficient education regarding importance of Hepatitis B vaccination. In this context, implementation of routine screening for vaccination can help identify unvaccinated individuals or those who need booster doses. Moreover, institutional policies regarding mandatory vaccination against Hepatitis B can enhance full vaccination coverage.

There are a few limitations associated with our study. Firstly, the sampling method that included only pre-clinical medical students of a single medical college possibly decreased the generalizability (external validity) of our findings. Additionally, there was possibility of recall and information bias while collecting data from the participants. However, we tried to minimize the bias by using short, simple and concise questionnaire. For future studies, we suggest to include using more standardized and validated questionnaire and use of specific cues and prompts that can help participants recall the information easily. We collected data related to vaccination status but could not confirm it using serological methods. Moreover, this study was conducted among female pre-clinical medical students only, which did not allow for sex-based comparison. Despite these shortcomings, our study has many strengths. We have adopted whole sampling technique which possibly decreased the risk of sampling error and selection bias along with increasing accuracy of our findings. This study is the first of its kind to be conducted among pre-clinical medical students in Bangladesh, and it highlights one of the most important aspects of medical care and education. In addition, our study signifies the need to raise awareness about Hepatitis B and the importance of HBV vaccination among medical students before the commencement of their clinical rotations.

## Conclusions

In present study, only one-third of the students were vaccinated against Hepatitis B, and only one-fifth of those vaccinated had received complete vaccinations. This is lower than other South East Asian countries, highlighting the vulnerability of medical students in Bangladesh to Hepatitis B infection. The most common reason for non-vaccination was lack of awareness of one's vaccination status. Additionally, only one-fourth of the students demonstrated overall good knowledge, attitude, and practice related to Hepatitis B. Therefore, it is crucial to provide education on Hepatitis B, its risk factors, and the importance of vaccination to pre-clinical medical students in Bangladesh.

## Author Contributions

**Conceptualization:** Ramesh Lamichhane.

**Data curation:** Aashika Rai, Pratikshya Ojha, Kripa Maharjan, Hamida Sultana Ruche, Madhusudan Saha.

**Formal analysis:** Ramesh Lamichhane, Pritha Adhikari, Bishnu Deep Pathak.

**Investigation:** Ramesh Lamichhane, Aashika Rai, Pratikshya Ojha, Kripa Maharjan.

**Methodology:** Ramesh Lamichhane, Pritha Adhikari, Madhusudan Saha.

**Project administration:** Aashika Rai, Pratikshya Ojha, Kripa Maharjan, Hamida Sultana Ruche, Madhusudan Saha.

**Supervision:** Ramesh Lamichhane, Madhusudan Saha.

**Writing – original draft:** Ramesh Lamichhane, Bishnu Deep Pathak.

**Writing – review & editing:** Pritha Adhikari, Bishnu Deep Pathak, Aashika Rai, Pratikshya Ojha, Kripa Maharjan, Hamida Sultana Ruche, Madhusudan Saha.

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
