## [Decision Letter · Decision Letter 0]

9 Jun 2024

PONE-D-24-19904Knowledge, Attitude, and Practice towards Hepatitis B and Vaccination Status of Pre-clinical Medical Students at Sylhet Women’s Medical College, BangladeshPLOS ONE

Dear Dr. Lamichhane,

Thank you for submitting your manuscript to PLOS ONE. After careful consideration, we feel that it has merit but does not fully meet PLOS ONE’s publication criteria as it currently stands. Therefore, we invite you to submit a revised version of the manuscript that addresses the points raised during the review process.

We look forward to receiving your revised manuscript.

Kind regards,

Dhan Bahadur Shrestha, MBBS

Academic Editor

PLOS ONE

Journal Requirements:

3. Please note that your Data Availability Statement is currently missing the DOI/accession number of each dataset or a direct link to access each database. If your manuscript is accepted for publication, you will be asked to provide these details on a very short timeline. We therefore suggest that you provide this information now, though we will not hold up the peer review process if you are unable.

4. Please upload a copy of Supporting Information which you refer to in your text on page 26. 

**Additional Editor Comments:**

I myself carefully read the manuscript and agreed to the comments by both reviewers, please revise the manuscript with needful considering those comments.

Reviewers' comments:

Reviewer's Responses to Questions

**Comments to the Author**

1. Is the manuscript technically sound, and do the data support the conclusions?

Reviewer #1: Yes

Reviewer #2: Partly

2. Has the statistical analysis been performed appropriately and rigorously? 

Reviewer #1: Yes

Reviewer #2: No

3. Have the authors made all data underlying the findings in their manuscript fully available?

Reviewer #1: Yes

Reviewer #2: Yes

4. Is the manuscript presented in an intelligible fashion and written in standard English?

Reviewer #1: Yes

Reviewer #2: No

5. Review Comments to the Author

Reviewer #1: In this study, the authors investigated the level of knowledge, attitude and practice regarding hepatitis B and vaccination among pre-clinical students in a medical college in Bangladesh in a descriptive cross-sectional study that included 189 participants. They conclude that only 1/3 of the participants was vaccinated and that lack of awareness of one's vaccination status was the most common reason. Only 1/4 of participants had good KAP related to hepatitis B. Vaccination status and previously received information was associated with high KAP score.

There is limited literature regarding KAP about hepatitis B and vaccination in medical students in Bangladesh. The methods are good and findings from the study have real-world implications as high KAP scores were associated with vaccination status. Thus, information dissemination in this population is crucial as only 1/4 population had good KAP score.

I would like to provide the following constructive feedback to the authors:

1. For questions- "vaccination against hepatitis B- yes or no" or "dose of HBV vaccine received- 1,2,3, 3 +booster?", was there an option indicating "i don't know" or an option to skip? This important as 43% of participants were "not aware of their hepatitis B vaccine status" so it is likely that 43% of the participants answered the 2 questions inaccurately. Recall bias is a big limitation of the study, which the authors do mention in the limitation section. They also mention that serological methods to confirm vaccination status would have been ideal to overcome this. However, I would recommend describing it more, including possible steps the authors took to minimize this bias in the study.

2. Was pre-testing/pilot testing performed? Some questions can possibly be interpreted in ways the authors did not indent/ not be clear (eg: Lack of accessibility for Hepatitis B vaccine, not aware of the Hepatitis B vaccine status). Also were any tests of validity of the questionnaire performed prior to the survey.

3. Authors did a good work with literature review in discussion but also describe the clinical/real-world implications of the results in the discussion.

4. Describe briefly the informed consent process- were participants told of what data collected and stored, who the investigators are, the purpose of study, etc. What personal information was collected?

5. Were any steps taken to prevent multiple entry of data from a single participant? How was it ensured that each entry of the survey was a unique participant?

6. Briefly mention what is the guideline for HBV vaccination in Bangladesh- childhood vaccination, healthcare workers? If it was part of childhood immunization program for more than 20 years, it is unlikely that 66% were not vaccinated (unless immunization coverage/ compliance is poor in the country)

7. I would add lack of generalizability/ ext validity in limitation- as study only included 1st and 2nd year medical students, and also 1 medical college in the country.

Minor points

- Line 280- sample size is not small as the entire population (1st and 2nd yr students at the medical college) was sampled.

minor

- line 82- this is not "non-probability convenient sampling method" as the whole population (all 1st and 2nd year students of the medical college) was used as sample.

- Table 1. 5th row, 3rd column- should be 21 (20-21)

- Table 5- present as n(%)

I would like to commend the effort of all the authors who put together an excellent study. I would recommend acceptance of the manuscript after minor revisions.

Reviewer #2: The authors have presented a work with findings from the perspective of their country. However, there are major points that should be corrected

- Background needs to better demonstrate the novelty of this resource, and add more information on how is hepatitis B vaccine given in their nation (for free or charge).

- Methods are unclear about recruitment and survey procedures.

- The sampling technique and sample size are not clear yet. Explain how the study size was arrived at

- Define all outcomes and give details of methods of assessment. In addition, the authors need to provide the cut-off level of good knowledge, attitude and practice, and references. For assessing KAP of the previous studies, knowledge, and practice variables are often classified as correct and incorrect, and attitude is assessed by using a five-point Likert scale and classified as positive and negative, the authors need to consider this. For KAP studies, the variable definition is not accurate, the results are less valid and reliable.

- The questionnaire contained 5 for knowledge as described in the methods, the authors can explain how to calculate the knowledge sum of 54.01±7.57 (table 3)

- The authors need to present the association between baseline characteristics and knowledge, attitude, and practice separately, not combining them (Table 5)

6. PLOS authors have the option to publish the peer review history of their article (what does this mean?). If published, this will include your full peer review and any attached files.

Reviewer #1: **Yes: **Sajog Kansakar

Reviewer #2: No

---

## [Author Response · Author response to Decision Letter 0]

16 Jun 2024

RESPONSE TO ACADEMIC EDITOR:

Thank you for this comment

We have edited as required in the manuscript.

2. Please provide additional details regarding participant consent. In the ethics statement in the Methods and online submission information, please ensure that you have specified (1) whether consent was informed and (2) what type you obtained (for instance, written or verbal, and if verbal, how it was documented and witnessed). 

Thank you for this comment 

Informed written consent was taken from each participant. We have edited as required in the manuscript.

3. Please note that your Data Availability Statement is currently missing the DOI/accession number of each dataset or a direct link to access each database. If your manuscript is accepted for publication, you will be asked to provide these details on a very short timeline. We therefore suggest that you provide this information now, though we will not hold up the peer review process if you are unable.

Thank you for this suggestion

We have added DOI link of figshare data in the manuscript as required.

4. Please upload a copy of Supporting Information which you refer to in your text on page 26. 

Thank you for this suggestion

We have uploaded the copy of supporting information as referred in the text (figshare doi link: https://doi.org/10.6084/m9.figshare.25845664.v1).

RESPONSE TO REVIEWER-1:

1. For questions- "vaccination against hepatitis B- yes or no" or "dose of HBV vaccine received- 1,2,3, 3 +booster?", was there an option indicating "i don't know" or an option to skip? This important as 43% of participants were "not aware of their hepatitis B vaccine status" so it is likely that 43% of the participants answered the 2 questions inaccurately. Recall bias is a big limitation of the study, which the authors do mention in the limitation section. They also mention that serological methods to confirm vaccination status would have been ideal to overcome this. However, I would recommend describing it more, including possible steps the authors took to minimize this bias in the study.

Thank you for these comments. Actually, the question “reason for not being vaccinated against Hepatitis B” was applicable for those who responded ‘NO’ to vaccination. And, this was the most commonly reported cause of non-vaccination among the participants.

We have tried to explain more on methods that can be used to reduce recall bias in this type of study. We have edited this in the DISCUSSION section of the manuscript.

2. Was pre-testing/pilot testing performed? Some questions can possibly be interpreted in ways the authors did not indent/ not be clear (eg: Lack of accessibility for Hepatitis B vaccine, not aware of the Hepatitis B vaccine status). Also were any tests of validity of the questionnaire performed prior to the survey?

Thank you for this comment. These questionnaires were adapted from Shrestha DB et. al, 2020 with permission. This study was conducted in Nepal with pre-testing done in 5% of the study sample. So, considering the similar population characteristics in South Asian region, we did not perform pre-testing in our sample.

However, we did assess internal consistency of our questionnaires that showed a good internal consistency, with a Cronbach's alpha of 0.71.

3. Authors did a good work with literature review in discussion but also describe the clinical/real-world implications of the results in the discussion.

Thank you for your compliment and comment. We have edited in the DISCUSSION section of the manuscript, where we have tried to describe clinical implications of our results.

4. Describe briefly the informed consent process- were participants told of what data collected and stored, who the investigators are, the purpose of study, etc. What personal information was collected?

Thank you for this comment. We have explained this information in detail in methods section in the manuscript.

5. Were any steps taken to prevent multiple entry of data from a single participant? How was it ensured that each entry of the survey was a unique participant?

Thank you for this concern. We had sent the google forms to each participant in their social media platform (facebook, whatsapp, email, etc). Each form was designed to accept only one unique response, preventing resubmission once finished. So, there was no possibility of multiple entry of data from a single participant.

6. Briefly mention what is the guideline for HBV vaccination in Bangladesh- childhood vaccination, healthcare workers? If it was part of childhood immunization program for more than 20 years, it is unlikely that 66% were not vaccinated (unless immunization coverage/ compliance is poor in the country)

Thank you for this comment. We have added this information in DISCUSSION section of the manuscript.

7. I would add lack of generalizability/ ext validity in limitation- as study only included 1st and 2nd year medical students, and also 1 medical college in the country.

Thank you for this comment. We totally agree with you that there is lack of generalizability/external validity in our study due to inclusion of pre-clinical medical students of one medical college only. We have added this in our manuscript.

8. Minor points

- Line 280- sample size is not small as the entire population (1st and 2nd yr students at the medical college) was sampled.

- line 82- this is not "non-probability convenient sampling method" as the whole population (all 1st and 2nd year students of the medical college) was used as sample.

- Table 1. 5th row, 3rd column- should be 21 (20-21)

- Table 5- present as n(%)

Thank you for these valuable suggestions. We have edited accordingly in the manuscript.

RESPONSE TO REVIEWER-2:

1. Background needs to better demonstrate the novelty of this resource, and add more information on how is hepatitis B vaccine given in their nation (for free or charge).

Thank you for this comment. 

Studies of this nature are commonly conducted worldwide. However, given that Bangladesh is situated in an intermediate endemic zone with lower Hepatitis B vaccination rates compared to other Southeast Asian nations, this research holds significant clinical and administrative importance. While the topic may lack novelty, it remains practically important from a public health perspective, as medical students represent a vulnerable population with respect to Hepatitis B infection.

We have added the information related to Hepatitis B vaccination in Bangladesh in DISCUSSION section in the manuscript.

2. Methods are unclear about recruitment and survey procedures.

Thank you for this comment

We adopted whole sampling technique, i.e. included all the pre-clinical medical students of a medical college. A questionnaire was sent to individual participant in the form of Google forms via different social media (facebook, whatsapp, email), and their responses were recorded in Excel form. The first page of the Google form consists of background and objectives of the study, authors’information and consent options. Those who consented were forwarded to next pages which consist of actual questionnaires. All these responses were accurately recorded subsequently in Excel form which was later used for data analysis and interpretation. We have tried to explain this in the manuscript in more elaborative way. Please kindly consider it.

3. The sampling technique and sample size are not clear yet. Explain how the study size was arrived at

Thank you for this comment

We used whole sampling technique which included all the pre-clinical medical students (MBBS 1st and 2nd academic years), i.e. Sample size = Total population

4. Define all outcomes and give details of methods of assessment. In addition, the authors need to provide the cut-off level of good knowledge, attitude and practice, and references. For assessing KAP of the previous studies, knowledge, and practice variables are often classified as correct and incorrect, and attitude is assessed by using a five-point Likert scale and classified as positive and negative, the authors need to consider this. For KAP studies, the variable definition is not accurate, the results are less valid and reliable.

Thank you for this comment and suggestions.

These questionnaires were adapted from Shrestha DB et. al, 2020 with permission.1 Similar pattern of scoring was used in our study as well. We have edited an explanation about KAP scoring and different variables in the manuscript itself. 

5. The questionnaire contained 5 for knowledge as described in the methods, the authors can explain how to calculate the knowledge sum of 54.01±7.57 (table 3)

Thank you for this suggestion.

We have explained this information in the manuscript. Actually, total score was calculated by adding individual score (based on Likert scale) on each item in knowledge, attitude and practice section.

6. The authors need to present the association between baseline characteristics and knowledge, attitude, and practice separately, not combining them (Table 5)

Thank you for this suggestion

Exploring association between baseline characteristics and knowledge, attitude and practice separately is a very good approach but it does not fall under our research objectives. Instead, we have studied the association between overall score (categorical) and baseline characteristics. It is because this association has been shown by a reference study by Shrestha DB et al,1 where association has been demonstrated between total KAP score category and baseline characteristics.

REFERENCE

1. Shrestha DB, Khadka M, Khadka M, Subedi P, Pokharel S, Thapa BB. Hepatitis B vaccination status and knowledge, attitude, and practice regarding Hepatitis B among preclinical medical students of a medical college in Nepal. PLoS One [Internet]. 2020 Nov 1 [cited 2024 Mar 12];15(11):e0242658. Available from: https://journals.plos.org/plosone/article?id=10.1371/journal.pone.0242658

---

## [Decision Letter · Decision Letter 1]

5 Aug 2024

PONE-D-24-19904R1Knowledge, Attitude, and Practice towards Hepatitis B and Vaccination Status of Pre-clinical Medical Students at Sylhet Women’s Medical College, BangladeshPLOS ONE

Dear Dr. Lamichhane,

Thank you for submitting your manuscript to PLOS ONE. After careful consideration, we feel that it has merit but does not fully meet PLOS ONE’s publication criteria as it currently stands. Therefore, we invite you to submit a revised version of the manuscript that addresses the points raised during the review process. As pointed out by reviewers, some methods and write up needs more clarity.

We look forward to receiving your revised manuscript.

Kind regards,

Dhan Bahadur Shrestha, MBBS

Academic Editor

PLOS ONE

Journal Requirements:

Reviewers' comments:

Reviewer's Responses to Questions

**Comments to the Author**

1. If the authors have adequately addressed your comments raised in a previous round of review and you feel that this manuscript is now acceptable for publication, you may indicate that here to bypass the “Comments to the Author” section, enter your conflict of interest statement in the “Confidential to Editor” section, and submit your "Accept" recommendation.

Reviewer #1: All comments have been addressed

Reviewer #3: (No Response)

2. Is the manuscript technically sound, and do the data support the conclusions?

Reviewer #1: Yes

Reviewer #3: Yes

3. Has the statistical analysis been performed appropriately and rigorously? 

Reviewer #1: Yes

Reviewer #3: Yes

4. Have the authors made all data underlying the findings in their manuscript fully available?

Reviewer #1: Yes

Reviewer #3: Yes

5. Is the manuscript presented in an intelligible fashion and written in standard English?

Reviewer #1: Yes

Reviewer #3: Yes

6. Review Comments to the Author

Reviewer #1: (No Response)

Reviewer #3: Minor: I recommend using the “pre-clinical medical student” throughout the manuscript to resolve the confusion associated with use of “pre-clinical medical student”, “medical student” or “health science students”.

Need grammatical corrections throughout the manuscript to make it easily comprehendible for the reader. You may take help of English language expert.

Abstract:

In materials and methods: “An online, single-center, descriptive cross-sectional study”: Use web-based rather than on “online”

In result: instead of only vaccine, mention the vaccine being Hepatitis B vaccine

Manuscript

Method:

- Reason for the choosing pre-clinical medical students not clear and also the risk of exposure to hepatitis B among participants not mentioned, like whether they have exposure to clinical practice or not, if yes, how’s the risk of the hepatitis B as per hospital record on needle injury among targeted participant group

- Reason for the selection of whole sampling method need to be explained

- Need to acknowledge about the participants not giving consent, whether they were included or excluded

Discussion:

- “Moreover, pre-clinical medical students are less experienced and frequently sustain needle-stick injuries while performing different procedures with infected patients.” Need reference for this line and whether it applies to participants of this study or not.

- Findings of study were compared to findings from other studies without mentioning whether they have similar cohort of participants or different.

7. PLOS authors have the option to publish the peer review history of their article (what does this mean?). If published, this will include your full peer review and any attached files.

Reviewer #1: No

Reviewer #3: No

---

## [Author Response · Author response to Decision Letter 1]

19 Aug 2024

RESPONSE TO REVIEWERS’COMMENTS:

1. I recommend using the “pre-clinical medical student” throughout the manuscript to resolve the confusion associated with use of “pre-clinical medical student”, “medical student” or “health science students”.

Thank you for the suggestion. We have edited in the manuscript as appropriate.

2. Need grammatical corrections throughout the manuscript to make it easily comprehendible for the reader. You may take help of English language expert. 

Thank you for the valuable suggestion. We have tried our best to edit the manuscript accordingly.

3. In materials and methods: “An online, single-center, descriptive cross-sectional study”: Use web-based rather than on “online”

Thank you for the suggestion. We have edited accordingly.

4. In result: instead of only vaccine, mention the vaccine being Hepatitis B vaccine

Thank you for the suggestion. We have edited accordingly.

5. Reason for the choosing pre-clinical medical students not clear and also the risk of exposure to hepatitis B among participants not mentioned, like whether they have exposure to clinical practice or not, if yes, how’s the risk of the hepatitis B as per hospital record on needle injury among targeted participant group.

Thank you for this comment. Our study sample consists of pre-clinical medical students who are moving into their clinical phase in near future. The pre-clinical medical students have little experience and during their clinical posting, they are apparently exposed to infected patients. Moreover, they have to get involved in different types of invasive procedures putting them at a high risk of needle-stick injuries. So, this group of students is at high risk of acquiring Hepatitis B infection. We have described these things in detail with relevant references in our INTRODUCTION and DISCUSSION section. And, we have made a few edits in the manuscript as appropriate. Please kindly consider it.

6. Reason for the selection of whole sampling method need to be explained

Thank you for this comment. In our study, we have included all the medical students in their pre-clinical academic years (i.e. whole sampling method). This means, we have included the entire study population (i.e. pre-clinical medical students of a medical college) in this research. It helps to decrease selection bias, sampling error, and increase the accuracy of our findings. We have added this information in DISCUSSION section last paragraph.

7. Need to acknowledge about the participants not giving consent, whether they were included or excluded

Thank you for this valuable suggestion. We have edited this information in METHODS section of the manuscript.

8. “Moreover, pre-clinical medical students are less experienced and frequently sustain needle-stick injuries while performing different procedures with infected patients.” Need reference for this line and whether it applies to participants of this study or not.

Thank you for this comment. We have cited references 4, 5, 12, 13 for all these information in paragraph 1 of DISCUSSION. Since this is an integrated information, we thought referencing as a whole would suit better. Please consider it.

9. Findings of study were compared to findings from other studies without mentioning whether they have similar cohort of participants or different. 

Thank you for this comment. We totally agree with your opinion regarding our study findings being compared with other studies without mentioning the cohort characteristics. Most of the time, we have tried to compare and contrast our results with the studies from South East Asia region where there are approximately similar socio-economic and educational characteristics. However, there were differences in sample size, sampling method, gender distribution, and other baseline characteristics. All those studies were observational studies (cross-sectional type), so we think that this difference should not matter much with our reporting in DISCUSSION section. Basically, we tried to compare our findings of vaccination rate and possible barriers to vaccination with other countries of the world (both developed and developing). Please kindly consider it.

---

## [Editor Report · Decision Letter 2]

1 Sep 2024

Knowledge, Attitude, and Practice towards Hepatitis B and Vaccination Status of Pre-clinical Medical Students at Sylhet Women’s Medical College, Bangladesh

PONE-D-24-19904R2

Dear Dr. Lamichhane,

We’re pleased to inform you that your manuscript has been judged scientifically suitable for publication and will be formally accepted for publication once it meets all outstanding technical requirements.

Kind regards,

Dhan Bahadur Shrestha, MBBS

Academic Editor

PLOS ONE

---

## [Editor Report · Acceptance letter]

22 Sep 2024

PONE-D-24-19904R2 

PLOS ONE

Dear Dr. Lamichhane, 

I'm pleased to inform you that your manuscript has been deemed suitable for publication in PLOS ONE. Congratulations! Your manuscript is now being handed over to our production team.

Kind regards, 

on behalf of

Dr. Dhan Bahadur Shrestha 

Academic Editor

PLOS ONE